# New Strategies for Macrophage Re-Education in Cancer: An Update

**DOI:** 10.3390/ijms25063414

**Published:** 2024-03-18

**Authors:** Nadia Lampiasi

**Affiliations:** Istituto per la Ricerca e l’Innovazione Biomedica IRIB, Consiglio Nazionale delle Ricerche, Via Ugo La Malfa 153, 90146 Palermo, Italy; nadia.lampiasi@irib.cnr.it; Tel.: +39-0916809513

**Keywords:** macrophage, molecular target, TAMs, CAFs, TME, cytokines, miRNAs, cancer immunology

## Abstract

The association between cancer and inflammation is well established. Chronic inflammation represents a fundamental step in the development and progression of some types of cancer. Tumors are composed of a heterogeneous population of infiltrating cells including macrophages, fibroblasts, lymphocytes, granulocytes, and mast cells, which respond to signals from the microenvironment and, in turn, produce cytokines, chemokines, transcription factors, receptors, and miRNAs. Recent data demonstrate that, in addition to classical (M1) and alternative (M2) macrophage subtypes, there are many intermediate subtypes that potentially play different roles in response to environmental stimuli. Tumors are infiltrated by macrophages called TAMs that mainly display an M2-like phenotype and tumor growth-permissive activities. There is a bidirectional interaction between tumor cells and tumor-infiltrating cells that determines macrophage polarization and ultimately tumor progression or regression. These complex interactions are still unclear but understanding them is fundamental for the development of new therapeutic strategies. Re-educating tumor-permissive macrophages into anti-tumor macrophages is a new focus of research. This review aims to analyze the most recent articles investigating the interplay between tumors, tumor-infiltrating cells, and TAMs, and the strategies for re-educating tumor-permissive macrophages.

## 1. Introduction

Macrophages are plastic cells able to differentiate into alternative subtypes that perform opposite functions. M0 are naïve macrophages that support the homeostasis of the organism. When homeostasis is lost, due to infection, disease, and cancer, M0 macrophages can differentiate into M1 classically activated or M2 alternatively activated phenotypes. M1 macrophages are pro-inflammatory phenotype and produce pro-inflammatory cytokines and chemokines. They support the physiological inflammatory process, which allows homeostasis to be restored. Unfortunately, sometimes the inflammatory process is prolonged over time, becomes chronic, or has a very strong intensity, and all this contributes to the establishment of pathological inflammation [1]. As a result of pathological inflammation, autoimmune diseases and some types of cancer can develop. The physiological inflammatory process is turned off by M2-type macrophages called the anti-inflammatory phenotype. These subtypes contribute with the cytokines produced to the restoration of homeostasis, and to the repair of wound healing. Unfortunately, in some cases, such as tumors, the contribution of this cell population is harmful to the organism, as it allows for the growth of the tumor and the formation of metastases [2,3]. The manipulation of macrophage subtypes suggests the possibility of intervening in the therapy of these diseases.

The formation of a solid tumor begins with the proliferation of a group of mutated cells that acquire new properties and escape the rules of normal cells. As the tumor mass grows, it acquires its own vessels and attracts cells from the blood circulation that can be useful for their growth [4]. The tumor cells produce soluble factors useful to affect the functions of lymphocytes, macrophages, and mast cells, essentially controlling the innate and adaptive immunity of the organism. M1-subtype macrophages exist in the tumor microenvironment (TME) and can potentially eliminate tumor cells by producing pro-inflammatory cytokines and by recruiting TH1 effector cells. However, the influence of various factors including local hypoxia, high levels of lactic acid, inflammation, and the resulting production of cytokines, chemokines, and ROS induce macrophages to develop pro-tumor properties [5,6]. In addition, the tumor enhances the transition of M0 and M1 macrophages towards the M2-like phenotype. In the early stages of tumor progression, M2-like macrophages attempt to repair tumor tissue through immunosuppression, tissue remodeling, and neovascularization. Unfortunately, tumor cells have learned to take advantage of these repair mechanisms and exploit these mechanisms for their benefit [7]. Indeed, pro-angiogenic cytokines/chemokines facilitate tumor cell proliferation, invasiveness, epithelial–mesenchymal transition (EMT), and metastasis formation [8,9]. In addition, immunosuppressive cytokines and immune checkpoint molecules expressed by macrophages, such as PD-L1, PD-L2, TIM-3, and VISTA, drive immunosuppression within the TME [5].

To complicate the situation, several M2 “hybrid” subtypes have recently been hypothesized, whose function is intermediate, i.e., they can simultaneously produce pro- and anti-inflammatory cytokines and molecules [10,11]. Indeed, M2 macrophages are a plethora of different cell subtypes (M2a, M2b, M2c, M2d) able to respond to multiple stimuli by producing cytokines and molecules both pro- and anti-inflammatory [12] (Table 1). 

M2a macrophages are responsible for removing cellular debris and repairing tissues through the release of pro-fibrotic factors such as fibronectin (FN), insulin-like growth factor (IGF), and transforming growth factor-β (TGF-β). In addition, M2a macrophages express high levels of STAT6 phosphorylated, produce anti-inflammatory cytokines as interleukin-(IL)-10, chemokine ligands (chemokine ligand-(CCL)-17, CCL-18, and CCL-22), and inhibit Th1 and Th2 response [13] (Table 1). Conversely, M2b and M2c subtypes are known as regulatory macrophages because they contribute to the maintenance or restoration of cellular homeostasis [14]. Indeed, M2b macrophages express high levels of ERK MAPK and AP-1 phosphorylated, secrete IL-10, tumor necrosis factor-alpha (TNF-α), and IL-1 [15], whereas M2c macrophages express high levels of STAT3 phosphorylated and secrete IL-10 and TGF-β, involved in tissue repair, matrix remodeling, and immunosuppressive behavior [14] (Table 1). Finally, the M2d subtype plays a role in wound healing through the expression of hypoxia-inducible factor 1-alpha (HIF-1α) and in angiogenesis through the production of vascular endothelial growth factor (VEGF) [16] (Table 1). In addition, M2d subtype releases both pro- and anti-inflammatory cytokines such as IL-10, TNF-α, and IL-12. This latter subtype has been identified in the ascites of patients with ovarian cancer and can promote tumor progression and has, therefore, been identified as TAMs [10].

The TME is complex and dynamic, composed of many different cell types including tumor cells (TCs), immune cells, cancer-associated fibroblasts (CAFs), and mesenchymal cells (Figure 1).

In the TME, all these cells can interact bidirectionally through the production and exchange of biological signals. At the molecular level, TME stimuli induce changes in the expression of transcription factors, receptors, or miRNA molecules of macrophages [10]. In turn, macrophages produce changes in the tumor cells themselves and in other cells that infiltrate the tumor. These complex interactions can be summarized in mathematical models [17].

All these phenotypes play a pivotal role in determining the final effects on the development of a tumor, i.e., whether a tumor will continue to grow or whether it will be eliminated. However, many aspects of macrophage polarization as TAMs, induced both by tumor cells and by TME, are still unclear and require further investigation and, for this reason, these interactions have become a research focus.

The aim of this review is to try to answer two important questions on the topic: (i) How do the microenvironment and tumor cells induce macrophage polarization towards a tumor-permissive subtype? (ii) Is it possible to re-educate these permissive macrophages towards an anti-tumor subtype?

## 2. Recruitment of Monocytes/Macrophages into Tumors

TAMs can be derived primarily from both resident macrophages and monocyte-derived macrophages, and, in addition, a smaller fraction is derived from myeloid-derived suppressor cells (MDSCs) that are closely related to blood monocytes [18,19,20]. Monocyte and macrophage infiltration within tumors is associated with a poor prognosis [21]. 

Monocytes are recruited into solid tumors through various mechanisms, including chemotaxis by cytokines, chemokines, and other cellular components.

Macrophage colony-stimulating factor (M-CSF), also known as CSF1, is essential for the survival and differentiation of myeloid precursors into macrophages. The CSF1/CSF1R axis drives the production and differentiation of monocytes circulating in the blood and tissue-resident macrophages and, therefore, plays a role in inflammatory processes, repair, and tissue homeostasis [22]. However, another relevant function of CSF1 is to promote M2-like polarization in tumors [23] and to act as a chemotactic factor for M2-like TAMs in the tumor microenvironment [24] (Table 2).

Chemokines are pivotal regulators of monocyte recruitment into tumors and can be produced by heterogeneous stromal cells in the TME activated by tumor cells, as well as by tumor cells themselves [25].

The main chemokine that affects the recruitment of monocyte/macrophages during the inflammatory process and cancer progression is C-C motif chemokine ligand 2 (CCL2), also known as monocyte chemoattractant protein 1 (MCP1) produced by injured and infected tissues, as well as by TCs (Table 2).

**Table 2 ijms-25-03414-t002:** Molecules involved in the recruitment of macrophages into tumor sites.

Molecule	Tumor	References
CSF1/CSFR1	CC, BC, OSCC	[23,24]
CCL2/MCP1	PC, EC, HCC, BC, PDA	[26,27,28,29,30]
CCL20	BC	[31]
CXCL1	Bladder cancer	[32]
CXCL12	Bladder cancer	[33]
CCL5/RANTES	Melanoma	[34]
RSK4	ESCC, OvTs, RCC	[35,36,37]

Legend: Colon cancer (CC), Breast cancer (BC), Oral squamous cell carcinoma (OSCC), Pancreatic carcinoma (PC), Esophageal carcinoma (EC), Hepatocellular carcinoma (HCC), Bladder cancer, Pancreatic ductal carcinoma (PDA), Esophageal squamous cell carcinoma (ESCC), Ovarian tumors (OvTs), and Renal cell carcinoma (RCC).

CCL2 binds to CCR2 but can bind to CCR4 and CCR5 [38], and can act as an antagonist on CCR3 [39]. Therefore, the biological role of CCL2 is multiple and complex and depends on the different expression of its receptors on macrophages in the tumor microenvironment.

CCR2 is predominantly expressed by monocytes/macrophages with strong pro-inflammatory functions triggering the release of pro-inflammatory cytokines, such as IL-1, IL-6, and TNF-α, but also tissue repair factors, such as VEGF, platelet-derived growth factor (PDGF) and TGF-β [40]. CCR2 acts in several ways to promote monocyte/macrophage recruitment. In fact, it contributes to the exit of monocytes from the bone marrow and is essential for the migration of monocytes from the blood to inflamed tissues [41,42]. However, CCR2-expressing monocytes/macrophages, particularly in tumor microenvironments, can be strongly immunosuppressive [43,44]. The CCL2-CCR2 axis has clinical significance in pancreatic tumorigenesis [26], esophageal carcinogenesis [27], hepatocellular carcinoma [28], and breast cancer [29,30]. This is why developing a CCR2 antagonist is a strategy to improve anti-tumor immunity.

Besides CCR2, CCR4 is a receptor expressed on monocytes/macrophages that promotes their migration [45,46]. The CCL2/CCR4 axis plays a role as a chemoattractant of monocytes/macrophages in pancreatic ductal carcinoma (PDA) and in murine pancreatic cancer [47,48,49]. Therefore, it is a potential immunological target to suppress TAM formation; in fact, CCR4 blockade improves prognosis in pancreatic cancer [48].

Other chemokines play a role in monocyte/macrophage recruitment. In particular, CCL20 expression on neutrophils and macrophages was also significantly correlated with immune cell infiltration in breast cancer, including monocytes, neutrophils, TAMs, Th1 cells, regulatory T cells, and exhausted T cells [31] (Table 2). Intriguingly, C-X-C motif chemokine 1 (CXCL1) has been identified to be increased in aggressive bladder cancer and correlated with TAM recruitment [32], a role also played by CXCL12 [33] (Table 2). In addition, the chemokine CCL5/RANTES, produced by T lymphocytes and macrophages, has a role in the macrophage’s recruitment at the melanoma tumor site, and interleukin-8 and interleukin-1β cytokines are involved in macrophage polarization [34] (Table 2).

Among other cellular components that may have a role in the recruitment of monocytes/macrophages, a study analyzed ribosomal s6 kinase 4 (RSK4) [35]. RSK4 is a serine/threonine kinase that belongs to the 90-kDa ribosomal S6 kinase family involved in the regulation of cell viability. It has been linked to poor prognosis and chemoresistance when overexpressed in patients with esophageal squamous cell carcinoma (ESCC), ovarian tumors, and renal cell carcinoma (RCC) [35,36,37] (Table 2). In addition, in human ESCC tissues, RSK4 was positively correlated with high infiltration of M0 and induction of M2-like polarization, suggesting that it may function as a target to re-educate macrophages [35].

## 3. Role of Monocytes/Macrophages in Cancer Progression and Metastasis

### 3.1. Crosstalk between Macrophages and CAFs

In the early stages of tumor development, naïve macrophages can acquire an M1-like phenotype and produce pro-inflammatory cytokines such as IL-12, which can activate and recruit anti-tumor cytotoxic T lymphocytes (CTLs). Subsequently, M1-like macrophages can be polarized towards an M2-like phenotype tumor permissive through cytokines/chemokines produced by CAFs present in the TME [50]. An interesting question is how do M1 macrophages and CAFs interact within the TME to promote cancer progression? Zainab and colleagues demonstrated that CAFs polarize M0 naïve and M1 macrophages towards an M2-like phenotype with a specific increase in CD200R and CD209 M2 markers [51] (Table 3). 

Naturally, the crosstalk between CAFs and macrophages occurs through the production of specific cytokines and chemokines such as CXCL13, CCL26, CXCL1, IL-1β, IL-21, IL-23, CCL3, MMP-1, RANKL, TNF-α, VEGF-D, CCL7, and SDF-1α [51].

In addition, these authors demonstrated that chemotherapy treatment (with Folfirinox) in pancreatic ductal adenocarcinoma (PDAC) determines a CAF-mediated strengthening of the M2 anti-tumor phenotype to the detriment of the M1 pro-inflammatory phenotype [51]. Many studies demonstrate that chemotherapy, but also radiotherapy, used to fight the tumor can instead promote resistance through the polarization of macrophages into M2-like TAMs [61].

Other interesting studies discovered that crosstalk between activated macrophages (am) M1 cells and CAF-like cells is implemented through the IL-6 production, which enhanced cancer stem cells (CSCs) growth and proliferation, supporting recurrence, metastasis, and angiogenesis in oral cancer progression [52,53] (Table 3). In addition, CSC growth and proliferation are sustained by TAM secretion of VEGF, which regulates macrophage polarization in triple-negative breast cancer (TNBC) [54]. On the other hand, the crosstalk between CSCs and TAMs is bi-directional. Indeed, CSCs with upregulation of CD24, CD47, and Intercellular Adhesion Molecule 1 (ICAM1) expression drive TAM polarization into a pro-tumorigenic HCC niche [55].

### 3.2. Crosstalk between Macrophage and Monocyte/Macrophage

The balance between the M1/M2 ratio is pivotal for tumor growth. Epithelial Membrane Protein 1 (EMP1) was identified to be the key gene indicative of tumor M1/M2 ratio, and higher EMP1 expression was associated with poor prognosis in many tumors [62,63]. Further analyses showed that EMP1 might promote tumor invasion and metastasis via EMT and focal adhesion (FA). Moreover, the expression level of EMP1 is related to immune infiltration including macrophages and neutrophils, and could serve as an indicator of immunotherapy efficacy [62].

The same naïve macrophages can attract monocytes as do, for example, alveolar macrophages (AMs) resident in the lung tissue. They recruit myeloid-derived monocytic suppressor cells (mo-MDSC), which, in turn, play a role in the formation of the premetastatic niche by increasing the expression of CCL12. CXCR3/TLR4 deficiency or inhibition impairs CCL12 expression in AMs and subsequent mo-MDSC recruitment to the premetastatic niche, thereby attenuating lung metastasis [56] (Table 3).

### 3.3. Crosstalk between TAMs and TME

Regarding TAMs influencing the activation status of TME cells, the cytokines C-X-C Motif Chemokine Ligand 9 (CXCL9), also known as monokine induced by gamma interferon (MIG), is the most studied. An elevated expression pattern of CXCL9 correlates with increased infiltration of B cells, macrophages, natural killer cells, and monocytes, and increased expression of immune checkpoint molecules and other CXCL family members, including CXCL10 and CXCL11 in TNBC [57] (Table 3). These findings confirm the regulatory role of CXCL9 in anti-tumor immunity and suggest its potential role in treatments to combat tumor growth. In addition, CXCL9 was used for prognostic clinical tumor outcomes together with Phosphoprotein 1 (SPP1 or osteopontin) in neck squamous cell carcinomas (HNSCCs). The polarity of CXCL9 and SPP1 in TAMs was positively associated with increased tumor infiltration of T cells, B cells, and dendritic cells (DCs), although CXCL9 positivity was associated with better prognostic value compared to SPP1. Indeed, macrophage positivity for CXCL9 or SPP1 defined anti- and pro-tumor macrophage phenotypes better than classical M1/M2 hallmarks expression [58]. These findings support the idea that M1 and M2 phenotypes are merely opposite ends of a broad phenotypic spectrum that reflects the complexity of TAMs in vivo.

### 3.4. Crosstalk between TAMs and TCs

To study the crosstalk between macrophages and TCs in vitro, two different approaches can be used. The first is a direct approach using a cell co-culture system, the second consists of producing a conditioned medium (CM), obtained during tumor cell culture, which is added to macrophage culture. The second approach is the most used because it allows us to distinguish who produces what. It has been shown that in the presence of CM derived from cancer cells, macrophages overexpressed markers of the M2-like phenotype including IL-10, VEGF, and MMP9 [64]. Therefore, tumor cells can regulate macrophage polarization independently of cell contact. Indeed, high mobility group protein B1 (HMGB1) in gastric cancer (GC) cell-derived exosomes induced M2-like macrophage polarization and promoted GC progression [59] (Table 3). Many studies confirm the importance of exosomes coming from cancer cells as a means of macrophage polarization. In NSCLC, A549-derived exosomes (A549-exo) induced M2 polarization of TAM through overexpression of Long noncoding RNAs (LncRNA) HOXC-AS2 and promoted the progression of NSCLC tumor [60] (Table 3). LncRNAs and miRNA may have a role in macrophage polarization [65,66]. Recent studies have shown that LncRNAs play an important role in the interaction between tumors and the tumor microenvironment [67]. Among them, LncRNA HOXC-AS2 is considered a novel tumor-associated LncRNA, and studies have shown that HOXC-AS2 may be related to glioma [68], gastric adenocarcinoma [69], digestive cancer [70], and non-small-cell lung cancer (NSCLC) [60]. 

## 4. Re-Education of TAM towards the Anti-Tumor Phenotype

TAMs play critical roles in tumor development such as promoting tumorigenesis, enhancing cancer progression and metastasis, inducing immunosuppression, resistance to chemotherapy, radiotherapy, and immunotherapy directed against immune system checkpoints [8,71,72]. Therefore, TAMs are the ideal therapeutic target to contrast cancer [73,74,75].

Two different strategies are used: TAM depletion and TAM reprogramming. The first strategy has been widely used, but the benefits for patients are limited or non-lasting [18,76,77]. Instead, the second strategy, which is based on macrophage plasticity, is useful and effective. Indeed, the plasticity of TAMs can be favorably directed to re-educate the M2 pro-tumor phenotype, which represents approximately 50% of the tumor mass, towards the M1 anti-tumor and pro-inflammatory phenotype, to enhance the patient’s anti-tumor response [77].

To re-educate macrophages, it is necessary to target the molecular nodes that specifically distinguish the two main subtypes M1 and M2 macrophages. Among others, we will address immune checkpoints, LncRNAs and miRNAs, signaling pathways, receptors, extracellular matrix components, and cytokines/chemokines (Table 4).

### 4.1. Immune Checkpoints

The immune checkpoints function like switches that must be activated (or deactivated) in order to initiate an immune response. However, cancer cells can employ these checkpoints to escape being targeted by the immune system. Expression of programmed cell death ligand 1 (PD-L1) can be found on both cancerous cells and tumor-infiltrating immune cells [78] (Table 4). PD-L1 expressed in tumor cells binds to PD-1 on T cells to counteract the effect of cytotoxic T cells and show resistance to the immune system (Figure 2A).

In addition, M2 TAMs can be immunosuppressive by inhibiting T-cell activity and enhancing the expression of PD-L1 in the TME cells. TAMs can block the function of PD-1/PD-L1 inhibitors by secreting anti-inflammatory cytokines and exosomes and increasing superficial immune checkpoint ligands in TME cells [79]. On the other hand, tumor cells may produce exosomes, which carry PD-L1, that can be phagocytized and expressed by macrophages (Figure 2B) [80]. High expression of PD-L1 in TAMs can inhibit immune activation, allowing tumor cells to escape the control of immune cells in oral squamous cell carcinoma (OSCC) [81]. Conversely, M1 macrophages downregulated PD-L1 expression in gastric cancer (GC) cells by loading miR-16-5p in exosomes and enhanced T-cell response (Figure 2C) [104]. Therefore, PD-L1 expression in macrophages may be a potential anti-tumor target, but little is known about the mechanism of PD-L1 expression in TAMs.

On the other hand, PD-1 expression on TAMs may participate to the osteosarcoma (OS) metastasis progression by inhibiting TAMs phagocytic activity [82] (Table 4) (Figure 2D). Therefore, PD-1 on TAMs may also be a potential anti-tumor target. Indeed, engineered exosomes loaded with siRNA for PD-1 and decorated with a peptide that specifically recognizes TAM M2-like macrophages are capable of re-educating macrophages towards M1-like subtype, restoring the immune activity of CD8+ T cells and remodeling the TME [83]. However, the selective blockade strategy of the PD-1/PD-L1 axis does not always have the desired effects since clinical studies showed patient resistance and tumor recurrence. Just as an example, in renal and bladder cancers, the PD-1 blockade strategy had no effect, but the combined treatment of anti-PD-1 with an original antisense oligonucleotide (ASO) for myeloid cell selective STAT3 knockdown (CpG-STAT3ASO) worked very well by reprogramming M2-like macrophages and promoting M1-like anti-tumor immunity in the TME, resulting in increased tumor growth control [88].

### 4.2. LncRNAs and miRNAs

LncRNAs and miRNAs can contribute to macrophage recruitment and polarization [65]. As discussed in the previous paragraph, NSCLC A549-derived exosomes (A549-exo) promoted M2 TAM polarization through overexpression of LncRNA HOXC-AS2. Therefore, HOXC-AS2 can be a target for macrophage re-education. Indeed, the downregulation of HOXC-AS2 in NSCLC promoted the M1-like polarization of TAM [60] (Table 4). The LncRNA NBR2 is an oncogene, but its function in colorectal cancer (CRC) is unclear. A recent study demonstrated that LncRNA NBR2 targets miR-19a in macrophages, and low levels of LncRNA NBR2 expression and overexpression of miR-19a promoted M2-like TAM polarization in CRC. In contrast, low levels of miR-19a expression promoted M1-like polarization, activating AMP and inhibiting HIF-1α and AKT/mTOR signaling pathways [66] (Table 4). The hypoxia created in the growing tumor regulates the activation of the cells infiltrating the tumor and the production of biological signals. The master regulator of hypoxia, the HIF-1α transcription factor, can increase miR-210-3p levels in lung adenocarcinoma and impair monocyte infiltration by inhibiting CCL2 expression. The inhibition of miR-210-3p promoted monocyte recruitment, but polarization towards an anti-tumor M1-like phenotype suggested a strategy for the re-education of M2-like macrophage [84] (Table 4).

### 4.3. JAK/STAT Pathway

The JAK2/STAT3 pathway is tumorigenic since it promotes M2-like polarization and, therefore, can be a target for macrophage re-education. Indeed, the inactivation of STAT3 and STAT6 transcriptional factors is associated with reducing tumor growth and metastasis in a model of breast and lung cancer [105,106,107]. Moreover, the inhibition of CCL18-mediated STAT3 phosphorylation, through immune responsive gene 1 (*IRG1*) overexpression, can inhibit macrophages M2-like polarization and impair the progression of intrahepatic cholangiocarcinoma (ICC) [89] (Table 4). The JAK2/STAT3 pathway is also activated by stem cells in GC through high secretion of IL-6/IL-8, and this contributes to M2-like TAM polarization and, consequently, promotes gastric cancer metastasis. Therefore, even in GC, the JAK2/STAT3 pathway should be considered a target to re-educate macrophages [22]. It has been reported that miRNA-214 and JAK2-inhibitor AG490 affected the JAK2/STAT3 signaling pathway and could prevent M2-like TAM polarization [85]. An interesting study conducted in NSCLC demonstrated that silencing miR-181b or using a JAK2 inhibitor can reprogram M2-like subtype macrophages [86], whereas miRNA-21 can reprogram M2-like macrophages in lung cancer [87]. These studies underline that silencing or increasing the expression of some miRNAs is functional to re-educate macrophages towards the anti-tumor subtype. Tripartite motif-containing 65 (TRIM65), which plays a crucial role in cancer progression [108,109], has E3 ubiquitin ligase activity and was found to directly bind NLRP3, promoting its ubiquitination in macrophages [110]. The TRIM65-JAK1/STAT1 axis is used by tumor cells to inhibit M1 macrophage polarization and promote tumor growth. Therefore, TRIM65 may be an anti-tumor target affecting the polarization of TAMs [90] (Table 4).

### 4.4. Receptors

Toll-like receptors (TLR) are receptors present on immune cells and can induce macrophage polarization [111]. A strategy to reprogram macrophages could employ the use of small molecules to target receptors, tyrosine kinases, or other transduction pathways in TAMs. Resiquimod (R848) is a TLR7/8 agonist, which contributes to the polarization of M1-like macrophages by activating the inflammasome, and, in turn, inducing a strong immune response [91] (Table 4). However, this drug administered systemically shows considerable toxicity [112], and, therefore, another delivery system must be developed. Some recent studies have demonstrated excellent efficacy in combining Resiquimod with nanoparticles loaded to target the TAMs present in the tumor. Fe-based metal–organic frameworks (MOFs) can release their cargo and iron in response to GSH in the TME [113]. Fan and colleagues designed MOF-based nanoparticles loaded with Caspase-1 for releasing their cargo in TME, and R848 for targeting M2-like TAM to induce re-education [92]. Anfray and colleagues developed polymeric nanocapsules (NCs) coated with hyaluronic acid (HA)–mannose to improve TAM targeting and loaded with a poly(I:C)+R848 combination able to induce M1-like macrophage polarization and cytotoxicity towards tumor cells [93] (Table 4). Zhao and colleagues devised a dual GSH- and pH-responsive nanoplatform encapsulated with doxorubicin (DOX) and R848 (GPNP) for combinatorial chemotherapy against tumor cells exhibiting drug resistance. GPNP released R848 into the TME, thereby reprogramming M2-like TAM into M1-like macrophages and releasing doxorubicin that killed MCF-7/ADR breast cancer cells [94].

Many other receptors have been investigated as possible targets for macrophage re-education. The triggering receptor expressed on myeloid cell 1 (TREM1) is an immunoreceptor expressed on neutrophils, monocytes/macrophages, and endothelial cells. It is overexpressed in HCC tissues and may serve as a tumor promoter in colorectal tumors [114], whereas its deficiency stimulates M1 pro-inflammatory macrophage polarization and impairs tumor growth [95] (Table 4). Regarding small molecules that inhibit kinases, the study conducted using SD-208, a transforming growth factor beta I receptor (TGF-βR1) kinase inhibitor, loaded on macrophage membrane-coated nanoparticles, demonstrated an inhibitory action on the polarization of M2-like macrophages and, therefore, a change in the tumor microenvironment from immunosuppressive (cold tumor) to immunostimulatory (hot tumor) [96].

### 4.5. Extracellular Matrix

Some components of the extracellular matrix can affect the recruitment of monocytes/macrophages [115,116]. Spondin 2 (SPON2), a matricellular protein 2, is essential for recruiting lymphocytes and initiating immune responses, as well as having a role in tumor progression [117,118]. Recent studies have shown that SPON2 expression is positively correlated with M2-TAM infiltration in clinical CRC tumors and with poor prognosis of patients. Blocking the SPON2/integrin β1/PYK2 axis impaired the migration of monocytes and cancer-promoting functions of TAMs in vivo [97] (Table 4). On the contrary, SPON2 in HCC TME promoted the infiltration of M1 macrophages and inhibited tumor metastasis [98]. Therefore, SPON2 exhibits context-dependent roles in cancer, and the heterogeneity of the microenvironment should always be considered.

Bone morphogenetic protein 2 (BMP2) is most significantly upregulated in breast cancer bone metastases where it activates macrophages. Therefore, BMP2 is a good target to reprogram macrophages in the early stages of tumor progression, before metastasis begins. In fact, subsequently, as the metastases progressed, the macrophages, even if reprogrammed in an anti-tumor sense, did not block the progression of the tumor [99] (Table 4). The same results are obtained in a lung metastasis breast cancer model in which BMP inhibition reduced the primary tumor [119].

### 4.6. Cytokines and Chemokines

Cytokines/chemokines are a group of molecules that significantly contribute to macrophage polarization and, therefore, represent an ideal target for reprogramming tumor macrophages. Some interesting studies on thyroid cancer have been conducted by Mazzoni and colleagues [100,101]. They have developed an in vitro model system for early and late tumor stages by using senescent thyrocytes and thyroid cell lines. CM from these cells polarized human monocytes as the M2-like phenotype, which exhibited tumor progression activity. Pharmacological inhibition of colony-stimulating factor 1 (CSF-1)/CSF-1R axis inhibited TAMs and impaired tumor progression [100] (Table 4). Moreover, the authors demonstrated in vivo, in a mouse transgenic model, the ability of senescent and tumor thyroid cells to recruit and polarize macrophages in the M2-like TAM pro-tumor phenotype and, in turn, TAM allow tumor progression [101]. Another very recent study demonstrated that the association of the CSF-1R inhibitor, PLX3397 (PLX), with the cytokine IL-12, which has the ability to stimulate the host’s immune activity, promoted repolarization of TAM, stimulated the proliferation and activation of T lymphocytes, reduced MDSCs recruitment, and suppressed tumor growth and metastasis [102] (Table 4). Cytokines/chemokines are largely produced by CAFs in the TME having a role in the crosstalk between macrophages and cancer cells, as stated above. CXCL12 is produced by CAFs in the TME and contributes to the secretion of plasminogen activator inhibitor-1 (PAI-1) by TAMs. PAI-1 acts as an inducer of tumorigenesis in HCC. Therefore, this factor may be a target for inhibiting the crosstalk between tumor cells, CAFs, and TAMs and impaired tumor growth [103].

## 5. Artificial Intelligence Can Help Re-Education of TAMs

The mathematical models can help precisely identify in the tumor the heterogeneity of macrophages, their spatial distribution, or the successful drug combination to educate macrophages. Many authors suggest that the best approach to fight cancer is [11] to focus on the re-education of a hybrid form of macrophages capable of producing cytokines and factors specific to the two phenotypes M1 and M2 in order to allow more effective and long-lasting effects [17,120,121,122,123].

The existence of many activated M1/M2 hybrid subtypes has been hypothesized using Boolean models or artificial intelligence [17,124,125], since macrophages can easily switch between states depending on the stimuli in TME and their spatial localization within the tumor [11]. Certainly, the activation of transcription factors such as NF-κB, AP-1, and STAT1 promote M1-like polarization or hybrid phenotypes with tumoricidal capacity, just as activation of HIF-1α and TGF-β promotes M2-like polarization or hybrid phenotypes with malignant capacity [126].

The spatial distribution of macrophages within the tumor conditions their polarization in relation to signals from the TME and interaction with other tumor and non-tumor cells. A new statistical function applied to an agent-based model (ABM), which takes into account the spatial distribution of macrophages within the tumor and the interaction with tumor cells, gives an idea of which macrophages should be targeted for re-education [127]. 

Quantitative systems pharmacology (QSP) modeling is an emerging mechanistic computational approach that can be developed in immuno-oncology to investigate the progression of disease mechanistically and quantitatively in response to various treatments. Chen Zhao and colleagues developed a large-scale QSP model for predicting temporal, dose-dependent, quantitative, and single-cell behaviors of M1/M2 macrophages in response to single and combinatorial signals [128]. This model, implemented with the dynamicity of TAMs in the TME, was used to predict the power of combination therapy in advanced TNBC [129]. Finally, the combination of QSP and ABM allows us to discover predictive biomarkers and develop combination therapies through in silico virtual trials as a consequence of TAM spatial heterogeneity distribution in the TME [130].

## 6. Conclusions

Immunotherapy has immense potential for treating cancer. Macrophage-targeted therapeutic approaches mainly aim to suppress the pro-tumor properties of M2-like TAM, to re-educate M2-like TAM towards the M1-type, or to activate the latter. All this can be achieved by using histone deacetylase and immune checkpoint inhibitors, Toll-like receptor agonists/antagonists, chemical compounds, miRNAs, siRNAs, or by silencing specific targets. Further studies are needed to define the scenario in which macrophages polarize in the tumor, as each patient and each tumor represent a unique challenge. Furthermore, the effects of chemotherapy and radiotherapy on macrophages very often induce an M2-like polarization, which can have a detrimental effect on the patient’s health. More recent studies agree that the winning strategy in the long term must be based on the possibility of re-educating pro-tumor macrophages towards the anti-tumor subtype.

New technologies that allow analysis of single cells, molecular profiling of different subtypes, and spatial localization within the tumor combined with artificial intelligence can quickly provide a useful platform for the development of new anti-tumor strategies (drugs, devices, etc.) that target macrophages.

## Figures and Tables

**Figure 1 ijms-25-03414-f001:**
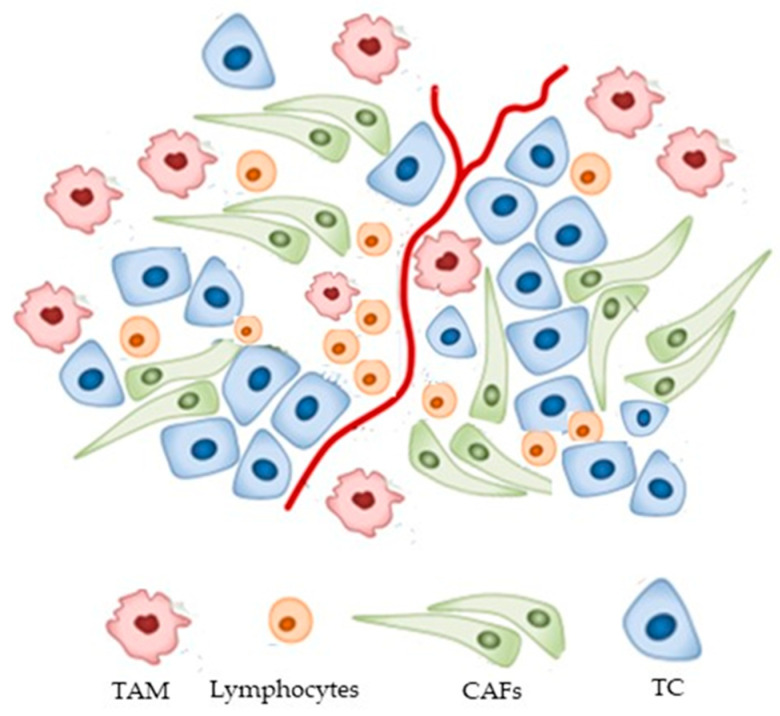
Schematic description of the TME. Abbreviations: tumor-associated macrophage (TAM), cancer-associated fibroblast (CAFs), tumor cells (TC).

**Figure 2 ijms-25-03414-f002:**
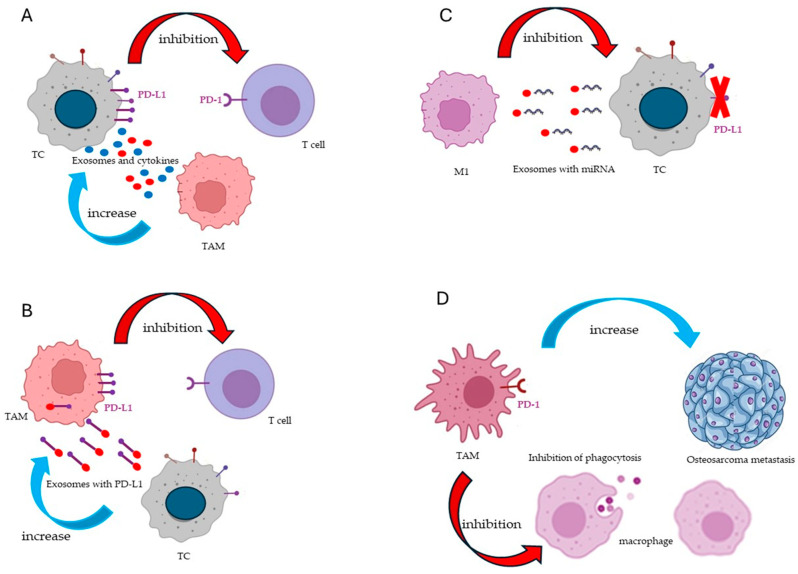
Immune checkpoints PD-L1 and PD-1 expression and function in macrophage and tumor cells (TC). (**A**) TAMs can indirectly inhibit immune activation by enhancing the PD-L1 expression on TCs through the release of exosomes and anti-inflammatory cytokines. PD-L1 expressed on TCs binds to PD-1 expressed on T cells and inhibits the cytotoxic effects of T cells. (**B**) TAMs can directly inhibit immune activation because they increase their expression of PD-L1 as a result of phagocytosis of PD-L1-carrying exosomes produced by TCs. (**C**) M1 macrophage can directly inhibit tumor growth by inhibiting PD-L1 expression in TCs through the release of exosomes carrying miR 16-5p. (**D**) PD-1-expressing TAMs inhibit macrophage phagocytosis and promote tumor growth and metastasis.

**Table 1 ijms-25-03414-t001:** Exemplification of the different subtypes of macrophages present in vivo and the main activated cytokines/chemokines and transcription factors.

Phenotype	Cytokines/Chemokines/TFs	References
M0	None	
M1	IL-6, TNF-α, IL-1β, NF-κB, AP-1,	[5]
M2	IL-10, TGF-β, IL-6	[5]
M2a	STAT6, JAK, FN, IGF, TGF-β, IL-10, CCL-17, CCL18, CCL22	[13]
M2b	AP-1, ERK, IRFs, IL-10, TNF-α, IL1	[14,15]
M2c	STAT3, IL-10, TGF-β	[14]
M2d	HIF-1α, VEGF, IL-10, TNF-α, IL-12	[16]

Legend: transcription factors (TFs), interleukin (IL), tumor necrosis factor-α (TNF-α), nuclear factor kappa-light-chain-enhancer of activated B cells (NF-kB), activator protein-1 (AP-1), transforming growth factor beta (TGF-β), signal transducer and activator of transcription (STAT), Janus kinase (JAK), fibronectin (FN), insulin-like growth factor-1 (IGF-1), chemokine (C-C) ligand (CCL), extracellular regulated kinase (ERK), interferon regulatory factors (IRFs), hypoxia-inducible factor 1α (HIF-1α), vascular endothelial growth factor (VEGF).

**Table 3 ijms-25-03414-t003:** Crosstalk between different phenotypes present in TME and molecules involved.

Phenotypes	Molecules Involved	References
M1-CAFs	CD200R, CD209	[51]
am(M1)-CAFs	IL-6	[52,53]
TAM-CSCs	VEGFA, ITGB3BP, ADAM9	[54,55]
CSCs-TAM	GAS6, ADAM9, ANXA1c	[55]
AM-moMDSC	CCL12	[56]
TAM-TME cells	CXCL9	[57,58]
TCs-TAM	HMGB1, LncRNA-HOXC-AS2	[59,60]

Legend: cancer-associated fibroblasts (CAFs), activated macrophage (am), cancer stem cells (CSCs), alveolar macrophage (AM), myeloid-derived monocytic suppressor cells (moMDSC), tumor cells (TCs), growth arrest-specific protein-6 (GAS6), ADAM Metallopeptidase Domain 9 (ADAM9), annexin A1c (ANXA1c), Integrin Subunit Beta 3 Binding Protein (ITGB3BP).

**Table 4 ijms-25-03414-t004:** Molecular nodes that can act as molecular targets to reprogram macrophages.

Nodes	Molecule Involved	References
Immune checkpoints	PDL-1/PD-1	[78,79,80,81,82,83]
LncRNAs	HOXC-AS2, NBR2	[60]
miRNAs	miRNA-19a, miRNA-210-3p, miR-181b, miRNA-214, miRNA-21	[66,84,85,86,87]
Signaling pathway	JAK/STAT	[22,85,86,88,89,90]
Receptors	TLR7/8, TREM1, TGF-βR1	[91,92,93,94,95,96]
Extracellular matrix	SPON2, BMP2	[97,98,99]
Cytokine	CSF-1/CSFR-1	[100,101,102]
Chemokine	CXCL12	[103]

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
