# Peer review of "New Strategies for Macrophage Re-Education in Cancer: An Update"

_ijms, 2024, doi:10.3390/ijms25063414_

Round 1

Reviewer 1 Report

Comments and Suggestions for Authors

This review provides a thorough summary of macrophage programming and its potential role in tumor progression and treatment. While the content is interesting, there are a few comments for the author to improve the manuscript.

1.        Line 73: Only a few abbreviations listed in table 1 are explained, which seems very odd.

2.        Line 78: Here it mentions that M2a produces chemokines such as CCL-17, CCL-18, CCL-22, however these were not listed in Table 1.

3.        Line 81: MAPKinase, typo.

4.        Line 125: It mentioned ovarian tumors here, however it never appeared in Table 2.

5.        Line 213: CCCC-X-C Motif Chemokine Ligand 9 (CXCL9), should be C-X-C Motif…

6.        Please check abbreviations. Sometimes the abbreviation and full names are repeated (e.g. GC in lines 284 and 324), sometimes abbreviations are listed but never used afterwards (TCs in line 229).

7.        Labeling of titles and subtitles are very confusing. Line 107 is #2, line 165 is #2.2, where is #2.1? Then it directly jumps from #2.3 to #5. Please check and correct.

8.        Line 427: Here the author mentioned Boolean-type computational models of macrophage polarization in terms of their utility in describing hybrid phenotypes, however such logic-based models tend to oversimplify things given their theoretical nature and often they lack temporal or quantitative features. The author should add discussion of mass-action based quantitative models of macrophage polarization as these models are usually more physiology-based and more validated in terms of descriptive/predictive power (e.g. PMID-33659877).

9.        Also, a quick search will return a quantitative systems pharmacology modeling study that investigated macrophage phenotype transition and its role in regulating immunotherapy in triple negative breast cancer.

10.    Table 5 lists several hypothesized hybrid phenotypes. I feel that such info should not be listed, as they are simulated by a theoretical model and has not been validated experimentally.

Comments on the Quality of English Language

Proofreading and writing style should be improved.

Author Response

This review provides a thorough summary of macrophage programming and its potential role in tumor progression and treatment. While the content is interesting, there are a few comments for the author to improve the manuscript.

I am very grateful to the reviewers for their useful comments that improved the manuscript.

  • Line 73: Only a few abbreviations listed in table 1 are explained, which seems very odd.
  • R: I added all the missing abbreviations.
  • Line 78: Here it mentions that M2a produces chemokines such as CCL-17, CCL-18, CCL-22, however these were not listed in Table 1.
  • R: I added the chemokines that were missing.
  • Line 81: MAPKinase, typo.
  • R: I corrected the typo error.
  • Line 125: It mentioned ovarian tumors here, however it never appeared in Table 2.
  • R: I corrected the omission.
  • Line 213: CCCC-X-C Motif Chemokine Ligand 9 (CXCL9), should be C-X-C Motif…
  • R: I corrected the typo error.
  • Please check abbreviations. Sometimes the abbreviation and full names are repeated (e.g. GC in lines 284 and 324), sometimes abbreviations are listed but never used afterwards (TCs in line 229).
  • R: I checked and corrected where necessary. TCs in the revised manuscript is cited.
  • Labeling of titles and subtitles are very confusing. Line 107 is #2, line 165 is #2.2, where is #2.1? Then it directly jumps from #2.3 to #5. Please check and correct.
  • R: I corrected the numbering of the paragraphs also following the suggestions of the second reviewer.
  • Line 427: Here the author mentioned Boolean-type computational models of macrophage polarization in terms of their utility in describing hybrid phenotypes, however such logic-based models tend to oversimplify things given their theoretical nature and often they lack temporal or quantitative features. The author should add discussion of mass-action based quantitative models of macrophage polarization as these models are usually more physiology-based and more validated in terms of descriptive/predictive power (e.g. PMID-33659877).
  • R: I followed the reviewer's suggestion and cited the PMID: 33659877 article. This part of the discussion has become a separate paragraph (as suggested by the second reviewer). Therefore, I decided to focus on the help that mathematical models can give to the study of the heterogeneity and distribution of macrophages within the tumor and implemented the discussion with other works. I hope that the changes made can make reading easier and more interesting this paragraph.
  • Also, a quick search will return a quantitative systems pharmacology modeling study that investigated macrophage phenotype transition and its role in regulating immunotherapy in triple negative breast cancer.
  • R: Thanks to the reviewer's suggestion I found one other work (Hanwen Wang 2022) which focus on polarized macrophages and the TNBC that I discussed.
  • Table 5 lists several hypothesized hybrid phenotypes. I feel that such info should not be listed, as they are simulated by a theoretical model and has not been validated experimentally.
  • R: In agreement with the suggestion, table 5 has been removed.

Reviewer 2 Report

Comments and Suggestions for Authors

The review article by Nadia Lampiasi discusses current achievements in the field of tumor macrophages. It presents recent advances in this topic that might be interesting for a wide audience of readers. However, the article reveals some limitations in the presentation of content, lacks proper structure and therefore it requires revision before publishing.

1.       Most significant limitation is that the manuscript does not contain any single illustration. Graphical representation of the text would make the reading flow better accepted. It would be beneficial to include illustrations for Tables 1, 2, and 3 to improve readability and comprehension.

2.       References in Tables are presented as names, but not in the format of numbers used in the text. Please correct.

3.        References in the tables. Consider including references only to original experimental papers in the tables, with a mixture of reviews if necessary.

4.       The chapter 2. Recruitment of monocytes/macrophages into tumors. I found this chapter is not only about the recruitment of monocytes but also a cross-talk between macrophages and other cells in the TME. Subdivide Chapter 2 into two or more sub-chapters: one focusing on the recruitment of monocytes/macrophages and the other(s) on the cross-talk between macrophages and other cells in the TME. Further divide the cross-talk sub-chapter into sections focusing on specific cell types like CAFs, CSCs, MDSCs, tumor cells, etc.

5.       Improve the readability of the chapter on the PD1-PDL1 axis, possibly by incorporating figures.

6.       2.3 Re-education of TAM towards the anti-tumor phenotype. Sub-divide this chapter on several with a focus on the pathway to target

7.       Consider summarizing available tested drugs for repolarizing macrophages in a table or figure, including information on clinical trials. It would be good to include a table with completed or ongoing clinical trials.

8.       The paragraph with mixed phenotype of THP1 and other macrophages is pretty messy. It does not fit into the title of this chapter on re-education. Create a separate chapter for discussing the hybrid phenotype of macrophages, as it does not fit well within the re-education chapter.

 Minor points

1.       Abstract. “Chronic inflammation represents a prerequisite for neoplastic transformation and tumor progression”. I am not sure that is "a prerequisite". Please re-phrase.

2.       Abstract. “Tumors are composed of a heterogeneous population of infiltrating cells including macrophages, fibroblasts, lymphocytes, granulocytes, and mast cells, which respond to signals from the microenvironment and, in turn, produce cytokines, chemokines, transcription factors, receptors, and miRNAs”.

I think, cancer cells are forgotten. Or perhaps not “tumors”, but rather the TME.

3.       Abstract. The transition to macrophages is not clear.

4.       “Unfortunately, tumor cells have learned to take advantage of these repair mechanisms and exploit these mechanisms to their advantage [7]”.

Two times “advantages”.

5.       “CCL2 binds to CCR2 but can bind to CCR4 and CCR5 [23], and can act as an antagonist on CCR3 [24]. Therefore, the biological role of CCL2 is multiple and complex and depend on the different expression of its receptors on macrophages in the tumor microenvironment.”

 What cells secrete CCL2? Specify the cell source of cytokines/chemokines like CCL2 and address similar issues for other molecules mentioned.

6.       Discuss the role of MSCF as an attractant for monocytes in the context of the manuscript's topic

Author Response

The review article by Nadia Lampiasi discusses current achievements in the field of tumor macrophages. It presents recent advances in this topic that might be interesting for a wide audience of readers. However, the article reveals some limitations in the presentation of content, lacks proper structure and therefore it requires revision before publishing.

I am very grateful to the reviewers for their useful comments that improved the manuscript.

  • Most significant limitation is that the manuscript does not contain any single illustration. Graphical representation of the text would make the reading flow better accepted. It would be beneficial to include illustrations for Tables 1, 2, and 3 to improve readability and comprehension.
  • R: As suggested by the reviewer I added two figures, one on the interaction between tumor and non-tumor cells in the tumor microenvironment, and the other on the interaction and functioning of the PDL-1/PD-1 checkpoint.
  • References in Tables are presented as names, but not in the format of numbers used in the text. Please correct.
  • R: The references were entered as a number.
  • References in the tables. Consider including references only to original experimental papers in the tables, with a mixture of reviews if necessary.
  • R: I edited according to the suggestion.
  • The chapter 2. Recruitment of monocytes/macrophages into tumors. I found this chapter is not only about the recruitment of monocytes but also a cross-talk between macrophages and other cells in the TME. Subdivide Chapter 2 into two or more sub-chapters: one focusing on the recruitment of monocytes/macrophages and the other(s) on the cross-talk between macrophages and other cells in the TME. Further divide the cross-talk sub-chapter into sections focusing on specific cell types like CAFs, CSCs, MDSCs, tumor cells, etc.
  • R: For a better understanding, paragraph 2.2 has become a separate paragraph numbered 3. And the latter has been divided into sub-paragraphs considering the different protagonists of the crosstalk as suggested.
  • Improve the readability of the chapter on the PD1-PDL1 axis, possibly by incorporating figures.
  • R: A figure is added in the manuscript (Figure 2).
  • 2.3 Re-education of TAM towards the anti-tumor phenotype. Sub-divide this chapter on several with a focus on the pathway to target
  • R: For a better understanding paragraph 2.3 has become a separate paragraph numbered 4. In accordance with the reviewer's suggestion this paragraph has been divided into sub-paragraphs.
  • Consider summarizing available tested drugs for repolarizing macrophages in a table or figure, including information on clinical trials. It would be good to include a table with completed or ongoing clinical trials.
  • R: Describing the pharmacological therapies currently used is outside the scope of the review. Furthermore, there are several very recent reviews published on the topic. (see Yuxin Lin 2019 doi.org/10.1186/s13045-019-0760-3, Zhao Huakan 2021 doi.org/10.1038/s41392-021-00658-5, Zhang Qjndong 2023, Qindong Zhang 2023 doi.org/10.3390/ijms24087493).
  • The paragraph with mixed phenotype of THP1 and other macrophages is pretty messy. It does not fit into the title of this chapter on re-education. Create a separate chapter for discussing the hybrid phenotype of macrophages, as it does not fit well within the re-education chapter.
  • R: I fully agree with the reviewer's suggestion therefore, I detached the topic from the previous paragraph. Moreover, in agreement with the second reviewer's suggestions, I expanded the discussion with suggested/new found manuscripts, and I eliminated table 5. I hope that the changes made can make reading easier and more interesting this paragraph.

 Minor points

  • “Chronic inflammation represents a prerequisite for neoplastic transformation and tumor progression”. I am not sure that is "a prerequisite". Please re-phrase.
  • R: Now the written sentence is like this. “The association between cancer and inflammation is well established. Chronic inflammation represents a fundamental step in the development and progression of some types of cancer”.
  • Abstract. “Tumors are composed of a heterogeneous population of infiltrating cells including macrophages, fibroblasts, lymphocytes, granulocytes, and mast cells, which respond to signals from the microenvironment and, in turn, produce cytokines, chemokines, transcription factors, receptors, and miRNAs”. I think, cancer cells are forgotten. Or perhaps not “tumors”, but rather the TME.
  • R: The cells that infiltrate the tumor refers to the other cells in addition to the tumor ones.
  • The transition to macrophages is not clear.
  • R: It's not clear to me what the reviewer means. However, polarization is better described in the introduction.
  • “Unfortunately, tumor cells have learned to take advantage of these repair mechanisms and exploit these mechanisms to their advantage [7]”.
  • Two times “advantages”.
  • R: The repetition has been eliminated.
  • “CCL2 binds to CCR2 but can bind to CCR4 and CCR5 [23], and can act as an antagonist on CCR3 [24]. Therefore, the biological role of CCL2 is multiple and complex and depend on the different expression of its receptors on macrophages in the tumor microenvironment.”
  •  What cells secrete CCL2? Specify the cell source of cytokines/chemokines like CCL2 and address similar issues for other molecules mentioned.
  • R: Cytokine/chemokine producing cells were added where necessary.
  • Discuss the role of MSCF as an attractant for monocytes in the context of the manuscript's topic
  • R: The role of M-CSF/M-CSFR has been discussed in the paragraph number 2, lines 127-133, and added in table 2.

Round 2

Reviewer 2 Report

Comments and Suggestions for Authors

The manuscript has been improved, however several issues need to be clarified before publishing.

1.       Figure one has lonely round T cells. However, the immune ell landscape is far beyond of only T cells. Why not to name them at least as lymphocytes (which actually includes T-, B- cells, NK cells)

2.       Figure 2 is sub-optimal. There is no description for A, B, C, D in legends, please correct. Cell-cell communication is illustrated only in the form of exosomes and it is missing soluble factors which is actually predominant component of intercellular communication. Please correct. In general, it is difficult to understand this figure and I would advice to improve it for better clarity.

3.       Chapter 5 – mathematical models. This title sounds grammatically incorrect. Why not using a term of artificial intelligence or mathematical modelling?

The manuscript has been improved; however, several issues need to be clarified before publishing.

  1. In Figure 1, there are only solitary round T cells depicted. However, the immune cell landscape extends far beyond T cells alone. Why not name them at least as lymphocytes, which actually includes T cells, B cells, and NK cells?
  2. Figure 2 is sub-optimal. There is no description for panels A, B, C, D in the legends; please correct this. Cell-cell communication is illustrated only in the form of exosomes, but it is missing soluble factors, which are actually a predominant component of intercellular communication. Please rectify this. In general, it is difficult to understand this figure, and I would advise improving it for better clarity.
  3. Chapter 5 – Mathematical Models: This title sounds grammatically incorrect. Why not use a term like artificial intelligence or mathematical modeling?

Author Response

Reviewer 2 second round

The manuscript has been improved, however several issues need to be clarified before publishing.

The reviewer is right. All suggested changes have been made.

  • Figure one has lonely round T cells. However, the immune cell landscape is far beyond of only T cells. Why not to name them at least as lymphocytes (which actually includes T-, B- cells, NK cells)

R:  Done. Now I have nominated the immune cells as lymphocytes.

2) Figure 2 is sub-optimal. There is no description for A, B, C, D in legends, please correct. Cell-cell communication is illustrated only in the form of exosomes and it is missing soluble factors which is actually predominant component of intercellular communication. Please correct. In general, it is difficult to understand this figure and I would advice to improve it for better clarity.

  1. Done. I have modified the figure and described the individual elements (A-D).

3) Chapter 5 – mathematical models. This title sounds grammatically incorrect. Why not using a term of artificial intelligence or mathematical modelling?

R: Done. Now the paragraph is titled “Artificial intelligence can help re-education of TAMs”
